# Towards a Better Understanding of Verapamil Release from Kollicoat SR:IR Coated Pellets Using Non-Invasive Analytical Tools

**DOI:** 10.3390/pharmaceutics13101723

**Published:** 2021-10-18

**Authors:** Julie Fahier, Branko Vukosavljevic, Laure De Kinder, Hugues Florin, Jean-François Goossens, Maike Windbergs, Florence Siepmann, Juergen Siepmann, Susanne Muschert

**Affiliations:** 1Univ. Lille, Inserm, CHU Lille, U1008, F-59000 Lille, France; julie.fahier@gmail.com (J.F.); lauredekinder@gmail.com (L.D.K.); hugues.florin@univ-lille.fr (H.F.); florence.siepmann@univ-lille.fr (F.S.); juergen.siepmann@univ-lille.fr (J.S.); 2Department of Drug Delivery, Helmholtz Center for Infection Research (HZI), Helmholtz Institute for Pharmaceutical Research Saarland (HIPS), 66123 Saarbruecken, Germany; brankovuk@hotmail.com (B.V.); windbergs@em.uni-frankfurt.de (M.W.); 3Department of Biopharmaceutics and Pharmaceutical Technology, Saarland University, 66123 Saarbruecken, Germany; 4Bayer AG, Chemical & Pharmaceutical Development, 42117 Wuppertal, Germany; 5ULR 7365-GRITA-Groupe de Recherche sur les Formes Injectables et les Technologies Associées, Faculté de Pharmacie, Université de Lille, CHU Lille, F-59000 Lille, France; jean-francois.goossens@univ-lille.fr; 6Institute of Pharmaceutical Technology and Buchmann Institute for Molecular Life Sciences, Goethe University Frankfurt, 60438 Frankfurt am Main, Germany

**Keywords:** controlled release, coated pellet, polyvinyl acetate, weakly basic drug, drug release mechanisms, film coating, non-invasive analytical tools

## Abstract

The aim of this study was to gain deeper insight into the mass transport mechanisms controlling drug release from polymer-coated pellets using non-invasive analytical tools. Pellet starter cores loaded with verapamil HCl (10% loading, 45% lactose, 45% microcrystalline cellulose) were prepared by extrusion/spheronization and coated with 5% Kollicoat SR:IR 95:5 or 10% Kollicoat SR:IR 90:10. Drug release was measured from ensembles of pellets as well as from single pellets upon exposure to acetate buffer pH = 3.5 and phosphate buffer pH = 7.4. The swelling of single pellets was observed by optical microscopy, while dynamic changes in the pH in the pellet cores were monitored by fluorescence spectroscopy. Also, mathematical modeling using a mechanistically realistic theory as well as SEM and Raman imaging were applied to elucidate whether drug release mainly occurs by diffusion through the intact film coatings or whether crack formation in the film coatings plays a role. Interestingly, fluorescence spectroscopy revealed that the pH within the pellet cores substantially differed upon exposure to acetate buffer pH = 3.5 and phosphate buffer pH = 7.4, resulting in significant differences in drug solubility (verapamil being a weak base) and faster drug release at lower pH: from ensembles of pellets and single pellets. The monitoring of drug release from and the swelling of single pellets indicated that crack formation in the film coatings likely plays a major role, irrespective of the Kollicoat SR:IR ratio/coating level. This was confirmed by mathematical modeling, SEM and Raman imaging. Importantly, the latter technique allowed also for non-invasive measurements, reducing the risk of artifact creation associated with sample cutting with a scalpel.

## 1. Introduction

Pellets are promising therapeutic formulations for oral administration since these small spherical multiparticulates offer multiple advantages compared to single unit dosage forms (e.g., tablets). The possibility to easily adjust individual doses and the convenience of swallowing are some of the reasons that improve patients’ compliance with these dosage forms, especially in children and the elderly [1,2,3,4]. However, drug release mechanisms from pellets coated with polymeric films can be rather complex. A variety of processes might be involved in the controlled release of the drug out of the device such as water penetration into system [5,6,7], drug dissolution [8,9], drug diffusion through the intact film coating [10,11], crack formation in the film coating due to hydrostatic pressure built up in the core [12,13], dynamic changes in the local pH in the pellet core [14] and in the composition of the film coating [15,16,17,18,19,20] (e.g., resulting from the leaching of a water-soluble compound into the surrounding bulk fluid, limited drug solubility effects, to mention just a few). It is desirable to know which mass transport mechanisms play a role in the coated pellet formulation of interest: The knowledge on the underlying drug release mechanisms contributes to an improved device optimization during product development as well as quality control. The elucidation of the drug release mechanisms from these multiple unit dosage forms can be challenging, since not only one mechanism might be of importance and since generally ensembles of pellets are observed (e.g., hundreds of pellets) at the same time, the measurements are therefore the sums of the behaviors of the various individual pellets, and the latter might substantially differ [16,21].

A variety of polymers and manufacturing processes are used to alter and control drug release from coated pellets [22,23,24,25,26,27]. The chemical composition of the polymeric membrane, its thickness and inner structure, but also the type of core material can have a major impact on the relative importance of the involved physico-chemical phenomena [19,28,29]. Different studies focus on the evaluation of the properties of the polymeric film coatings using a variety of in-, on- or off-line tools [30,31,32]. Polymeric film coatings can be either applied from organic solutions or aqueous dispersions [33,34,35,36]. Also, frequently, different types of polymers are blended to obtain specific film coating properties and drug release patterns [14,16,18,20,28,37]. Kollicoat SR 30D is an aqueous dispersion of 30% solid particles of polyvinyl acetate, also containing small amounts of poly(vinyl pyrrolidone) and sodium lauryl sulfate. Kollicoat IR is a water-soluble polyvinyl alcohol-polyethylene glycol graft copolymer. In this study, the two polymers were blended at different ratios and used to coat verapamil HCl loaded starter cores, which contained 10% (*w/w*) drug, 45% lactose and 45% microcrystalline cellulose. It has to be pointed out that also the solubility of the drug can be of major importance for the rate of drug release, since only dissolved drug is able to diffuse across an intact polymeric membrane. The drug investigated in this study, verapamil HCl, exhibits a strong pH dependent drug solubility, being freely water-soluble at low pH and only very slightly soluble at neutral pH [38]. In the case of such drugs (exhibiting pH dependent solubility), it is very interesting to monitor the local pH within the pellet cores during drug release. Eisenaecher et al. reported interesting results applying different methods to trace the micro environmental pH changes within multilayer tablets during dissolution, including the use of pH indicator dyes and fluorescence spectroscopy [39]. The use of pH indicators required destructive cross sectioning of the tablets, as also employed by Streubel et al. [38]. The cutting of the samples might be challenging, in particular in the case of coated pellets after exposure to the release medium, since the pellet might be mechanically fragile and the coating might detach. Cutts et al. [40] reported the traceability of the pH generated within uncoated minocycline pellets (a commercially available product) using confocal laser scanning microscopy. In this case, the coated pellets could not be monitored using this technique, since the coatings were opaque. In the present study, transparent polymeric coatings were applied, free of plasticizers and pigments or anti-tacking agents, in order to allow for pH monitoring in the pellet cores using fluorescence spectroscopy. Furthermore, Raman spectroscopy was used to visualize the film coatings as well as the pellet cores [41]. Importantly, Raman spectroscopy can be applied to analyze wet samples and can be used for chemically selective imaging, if the Raman spectra of the molecules of interest are sufficiently distinct.

## 2. Materials and Methods

### 2.1. Materials

Verapamil hydrochloride (HCl, Safic Alcan, Puteaux, France); microcrystalline cellulose (MCC, Avicel PH 101, FMC BioPolymer, Brussels, Belgium); lactose monohydrate (Lactochem fine powder, DFE Pharma, Goch, Germany); Kollicoat SR 30D (a 30% aqueous dispersion of polyvinyl acetate, also containing small amounts of poly(vinyl pyrrolidone) and sodium lauryl sulfate) and Kollicoat IR (polyvinyl alcohol-polyethylene glycol graft copolymer) (BASF, Ludwigshafen, Germany); two analogues of fluorescein: Anionic Oregon Green 488 dextran (10,000 MW, Invitrogen, Carlsbad, CA, USA) and anionic 2′,7′-*bis*-(2-carboxyethyle)-5-(and-6)-carboxyfluorescein (BCECF) dextran (10,000 MW, Invitrogen). Sodium acetate, acetic acid glacial, potassium dihydrogen phosphate, hydrochloric acid and sodium hydroxide (pellets, Fisher Scientific, Illkirch, France).

### 2.2. Solubility Measurements

The solubility of verapamil HCl in hydrochloric acid (0.1 N), acetate buffer pH = 3.5 (prepared by adjusting the pH of 800 mL of a 0.064 M sodium acetate solution with glacial acetic acid and filled up to 1.0 L) and phosphate buffer pH = 7.4 (United States Pharmacopeia, USP 40) was determined by adding excess amounts of drug to 50 mL of the aqueous media in brown glass flasks agitated at 80 rpm at 37 °C (GFL 3033, Gesellschaft fuer Labortechnik, Burgwedel, Germany). The pH was optionally adjusted with a 30% sodium hydroxide solution or 32% hydrochloride acid to 1.22, 3.3, 5.5, 6.25, 7.15, 7.5, 7.98, 8.68 and 9.31. Samples were taken from the supernatant, filtered and analyzed spectrophotometrically (UV 1650 PC, Shimadzu, Champs sur Marne, France, λ = 232 nm). In all cases, plateau values were observed. The experiments have been conducted in triplicate.

### 2.3. Preparation of Thin, Free Films

Thin free polymeric films were prepared using a spray gun, equipped with a 2 mm nozzle (LacAir SW, gravity spray gun, Lacme, La Fleche, France) and Teflon plates. The coating dispersions were prepared by dissolving appropriate amounts of Kollicoat IR in demineralized water and addition of Kollicoat SR 30D to attain 95:5 or 90:10 (*w:w*) polymer:polymer blends, respectively. The spray rate was set to 2 g/min and the atomization pressure was at 1 bar. The films were sprayed under continuous drying with a hair dryer and subsequently cured for 48 h at 60 °C in an oven. Verapamil HCl-loaded films were prepared accordingly, adding 1% (*w/w*) verapamil hydrochloride to the polymer dispersion.

### 2.4. Preparation of Coated Pellets

The starter cores were prepared by mixing 10% verapamil HCl, 45% lactose, 45% MCC (all percentages are in *w/w*) and in a plowshare blender (Loedige M20 Mixer, Gebrueder Loedige Maschinenbau, Paderborn, Germany) for 10 min. 600 mL demineralized water per kilogram of powder blend was progressively added in small portions. Optionally, fluorescent markers were subsequently added (0.025% of Oregon Green or 0.01% of BCECF dextran) using a planetary mixer (K-blade Kenwood Chef Titanium, De’longhi Kenwood, Clichy, France). The wetted mass was transferred into a counter rotating cylinder extruder with a perforated roll (diameter of perforations: 1.0 mm, wall-thickness: 3 mm, 32 rpm rotation speed; SKM/R, Alexanderwerk, Remscheid, Germany). The cylindrical extrudates (approximately 4 mm in length) were subsequently spheronized for 30 s at 765 rpm (Spheronizer model 15, Calveva, Dorset, UK). Batch sizes were of 1 kg, only in the case of pellets loaded additionally with fluorescence marker 150 g batches have been produced.

The pellet starter cores were coated in a bottom-spray fluid bed coater, equipped with a Wurster insert (Strea 1, Aeromatic-Fielder, Bubendorf, Switzerland). The coating dispersions (prepared as for the preparation of free films, described in the Section 2.3) were agitated using a magnetic stirrer at 450 rpm for 30 min prior to coating. The coating process parameters were as follows: 40 °C inlet temperature (resulting in a product temperature of around 37 °C), 1.2 bar atomization pressure, 110 m^3^/h fluidizing air flow, a spray rate of 2.5 g/min and using a nozzle of 1.2 mm in diameter. The pellets were coated until a weight gain of 5% or 10% (*w/w*) was achieved in the case of Kollicoat SR: Kollicoat IR 95:5 or 90:10 blends, respectively. The obtained pellets were oven cured at 60 °C for 48 h.

### 2.5. In Vitro Drug Release and Swelling Studies

In vitro drug release from ensemble of pellets (500 mg) was determined in 500 mL acetate buffer pH = 3.5 (prepared as described above) or phosphate buffer pH = 7.4 (USP 40) using the USP paddle apparatus (USP 40, 100 rpm, 37 °C). At predetermined time points, samples were withdrawn, filtered (through a filter needle) and analyzed for their drug contents by UV spectrophotometry (Shimadzu 1650, Champs-sur-Marne, France) at λ = 278.8 nm. All experiments were performed in triplicate, mean values ± standard deviations are reported.

Drug release kinetics from single pellets were determined in 400 µL release medium in well plates, agitated at 120 rpm at 37 °C (VarioSkan Flash, Thermo Scientific, Courtaboeuf, France). At predetermined time points, the entire bulk fluid was replaced with pre-heated, fresh medium. The samples were filtered and their drug content determined by UV spectrophotometry in a micro-cuvette (Shimadzu 1800, λ = 278.8 nm).

Dynamic changes in the diameter of single pellets were studied by placing individual pellets in small flasks containing 500 µL release medium. At predetermined time points, pictures were taken with an optical imaging system (Nikon SMZ-U, Nikon Tokyo, Japan; AxioCam1 and AxioVision v4.8, Carl Zeiss S.A.S., Marly, France).

### 2.6. Determination of the Partition Coefficient of the Drug

Drug free films (3 × 3 cm, thickness about 40 µm) were placed in flasks, filled with 100 mL acetate buffer pH = 3.5 (prepared as described above) containing 30 or 45 mg/mL verapamil hydrochloride. The flasks were horizontally shaken at 80 rpm at 37 °C (GFL 3033). Film samples were withdrawn (up to 7 d, until an equilibrium was reached) and dissolved in an ethanol:isopropyl alcohol:aqueous hydrochloric acid (32% *v/v*) blend (70:15:15, *v:v:v*), and the drug amounts were determined by UV spectrophotometry (Shimadzu 1800, λ = 290 nm). In addition, the drug content of the bulk fluids was determined by UV spectrophotometry (Shimadzu 1800, λ = 278.8 nm). All experiments were performed in triplicate, mean values ± standard deviations are reported.

### 2.7. Determination of the Diffusion Coefficient of the Drug

Thin, free films (5 × 5 cm, thickness about 40 µm) loaded with 1% verapamil HCl were exposed to 100 mL acetate buffer pH = 3.5 (prepared as described above) in flasks, which were horizontally shaken at 80 rpm at 37 °C (GFL 3033). Sink conditions were maintained throughout the experiments. At predetermined time points, 3 mL samples were withdrawn (replaced with fresh, pre-heated release medium) and analyzed for their drug content by UV-spectrophotometry (Shimadzu 1650, λ = 278.8 nm). All experiments were performed in triplicate, mean values ± standard deviations are reported. The following solution of Fick’s second law of diffusion was fitted to these experimentally determined drug release kinetics [42]:(1)$$\frac{M_{t}}{M_{infinity}}=1-\frac{8}{\pi^{2}}\overset{\infty }{\underset{n=0}{\sum }}\frac{1}{{\left(2n+1\right)}^{2}}exp\left(\frac{-D{\left(2n+1\right)}^{2}\pi^{2}t}{L^{2}}\right)$$
where, *M_t_* and *M_infinity_* designate the cumulative absolute amounts of drug released at time *t* and *infinity*, respectively; *D* is the apparent diffusion coefficient of the drug in the film; *L* denotes the thickness of the film.

This equation can be used, if the following conditions are given: (i) Drug diffusion is the dominant mass transport mechanism. (ii) The drug is initially homogenously distributed throughout the film. (iii) The drug is molecularly dispersed within the polymeric network (“monolithic solution”). (iv) Perfect sink conditions are provided throughout the experiment. (v) The geometry of the system is that of a thin, free film. (vi) The dimensions of the film do not change during the experiment (e.g., no noteworthy film swelling or dissolution).

### 2.8. Monitoring of the pH within the Pellets during Drug Release

Single pellets were treated as for the in vitro drug release measurements described above. Dynamic changes in the pH within the pellets were monitored non-invasively by fluorescence spectroscopy. The fluorescent markers Oregon Green (0.025%) or BCECF dextran (0.01%) were incorporated into the pellet starter cores. At pre-determined time points, the excitation spectrum of Oregon Green (λ = 400 to 510 nm) and BCECF dextran (λ = 400 to 520 nm) were traced with constant emission wavelengths of 526 nm and 534 nm respectively. The fluorescence intensity was expressed as the ratio of the measured intensity at the isosbestic wavelength (Oregon Green: 434 nm, BCECF: 454 nm) and the maximum wavelength (Oregon Green: 496 nm, BCECF: 506 nm). Polynomial standard curves were established with fluorescence marker solutions in several buffer media in the presence of placebo pellets. Four calibration curves were used to determine the pH from the measured fluorescence intensity ratios:

For Oregon Green-loaded pellets coated with 5% Kollicoat SR:IR 95:5:fluorescence intensity ratio = 0.0192 pH^3^ − 0.1414 pH^2^ − 0.1135 pH + 1.8282,(2)
and coated with 10% Kollicoat SR:IR 90:10:fluorescence intensity ratio = 0.0047 pH^3^ + 0.0377 pH^2^ − 0.8406 pH + 2.8056.(3)For BCECF-loaded pellets coated with 5% Kollicoat SR:IR 95:5:fluorescence intensity ratio = 0.0562 pH^3^ − 1.0257 pH^2^ + 5.6931 pH − 8.4905,(4)
and coated with 10% Kollicoat SR:IR 90:10:fluorescence intensity ratio = −0.0509 pH^3^ + 1.2425 pH^2^ − 10.119 pH + 27676.(5)

### 2.9. Scanning Electron Microscopy (SEM)

The morphology of the coated pellets’ surface before and after exposure to the release medium was observed using a scanning electron microscope (S-4700 Hitachi High-Technologies Europe, Krefeld, Germany). Prior to imaging, the samples were dried (at room temperature) and covered with a fine carbon layer under vacuum.

### 2.10. Confocal Raman Microscopy

Confocal Raman microscopy measurements were performed using a WITec alpha 300R+ (WITec, Ulm, Germany). Pellets were bisected using a scalpel and fixed on glass slides. The excitation source was a diode laser with a wavelength of 532 nm with a power of 10–40 mW before the objective. A confocal pinhole of 50 μm rejected signals from out-of-focus regions. Single spectra and line scans were recorded using a 50× objective (Epiplan Neofluar, Zeiss) with a numeric aperture (N.A.) of 0.8, while for the imaging of pellets 10× (N.A. 0.25) and 50× long distance (N.A. 0.55) magnification objectives were used. The lateral resolution was 1.1 µm for the 10× objective, and 0.5 µm for the 50× objective, respectively. The axial resolution was 3.1 µm when using the 50× objective. Raman spectra of the pure components were acquired with an integration time of 0.5 s and 10 accumulations; although image scans of the samples were recorded with integration time from 0.1 s to 0.5 s every 2–10 μm along the x- and y-axes and 0.5 s every 1 µm along the z-axis. All spectra were background subtracted, normalized to the most intense peak and converted into false-color images using WITec Project Plus software (WITec GmbH, Ulm, Germany). In these images, verapamil hydrochloride is depicted in red, pellet matrix (MCC and lactose) in blue, and the film coating (Kollicoat IR and Kollicoat SR) in yellow, respectively.

## 3. Results and Discussion

### 3.1. Drug Release from Ensemble of Pellets

Figure 1 shows the drug release kinetics from ensembles of pellets coated with 5% Kollicoat SR:IR 95:05 (a), or 10% Kollicoat SR:IR 90:10 (b) in acetate buffer pH = 3.5 and phosphate buffer pH = 7.4, respectively. The starter cores consisted of 10% verapamil HCl, 45% lactose, and 45% MCC. In all cases, sink conditions were provided in the well stirred release medium throughout the experiments. As it can be seen, verapamil release was faster in acetate buffer pH = 3.5 than in phosphate buffer pH = 7.4, irrespective of the type of coating. This might eventually be attributable to the fact that the solubility of verapamil (being a weak base) is strongly pH dependent: Figure 2 shows the dependence of the solubility of the drug as a function of the pH of the bulk fluid (acetate buffer pH = 3.5 and phosphate buffer pH = 7.4 were used, the pH was optionally adjusted with sodium hydroxide or HCl) at 37 °C. The results are in rather good agreement with data reported in the literature [38]. Looking at Figure 1a,b, it can be seen that verapamil release was slightly faster from pellets coated with only 5% Kollicoat SR:IR 95:05 compared to 10% Kollicoat SR:IR 90:10, irrespective of the type of release medium. This can be explained by the higher hydrophilicity and water-solubility of Kollicoat IR compared to Kollicoat SR, resulting in more permeable Kollicoat SR:IR 90:10 compared to Kollicoat SR:IR 95:05 coatings. The increase in coating thickness does not fully compensate for this difference in drug permeability. Thicker film coatings prolong the on-set of drug release, since the water penetration is slowed down [7].

In all cases the hypothesis is the following: Upon exposure of the pellets to acetate buffer pH = 3.5 and phosphate buffer pH = 7.4, the bulk fluids penetrate into the systems and dissolve the drug located in the cores. Depending on the type of release medium, the local pH within the pellet cores is different: lower in the case of acetate buffer and more neutral in the case of phosphate buffer. 

If this is true, the solubility in the pellet cores is higher upon exposure to the acetate buffer, explaining the observed higher drug release rates (assuming similar coating properties upon exposure to both media, in particular thickness and permeability). To evaluate the validity of this hypothesis, the pH within the pellet cores as well as potential dynamic changes thereof during drug release were non-invasively monitored by fluorescence spectroscopy.

Furthermore, it has to be pointed out that the drug release patterns shown in Figure 1 were observed with ensembles of pellets. The illustrated curves are the sums of hundreds of individual drug release patterns from single pellets. It might be that the single pellets behave similarly, but it might also be that their release profiles fundamentally differ [17,43]. For example, the Axelsson group showed that the relatively constant release rate observed from ensembles of remoxipride-loaded pellets coated with ethyl cellulose was the superposition of highly variable release profiles from single individual pellets [44]. This shows the necessity that, when aiming to obtain deeper insight into the underlying drug release mechanisms from coated pellets, also drug release from single pellets should be monitored.

### 3.2. Single Pellets: Drug Release, Core pH and Swelling

Figure 3 and Figure 4 show the experimentally measured verapamil release kinetics (blue curves, left y-axes) and dynamic changes in the local pH within the pellet cores (orange crosses, right y-axes) upon exposure to acetate buffer pH = 3.5 and phosphate buffer pH = 7.4 (left and right columns). The pellets were coated with 5% Kollicoat SR:IR 95:05 (Figure 3), or 10% Kollicoat SR:IR 90:10 (Figure 4). Single pellets were exposed to the release media in well plates under agitation at 37 °C. The fluorescence markers Oregon green and BCECF dextran were incorporated into the systems’ cores to allow for pH measurements by fluorescence spectroscopy through the transparent film coatings upon exposure to acetate buffer pH = 3.5 and phosphate buffer pH = 7.4, respectively.

Interestingly, the local pH within the pellet cores was indeed much lower upon exposure to acetate buffer pH = 3.5 than upon exposure to phosphate buffer pH = 7.4, irrespective of the type of film coating (Figure 3 and Figure 4). This is consistent experimental evidence for the hypothesis described above that pH differences in the pellet cores are playing a key role for the control of drug release, determining the solubility of verapamil in the systems’ cores. Also, please note that large parts of the single pellet release curves observed are about linear: This might indicate that saturated drug solutions eventually exist within the pellet cores during prolonged periods of time (the amount of water likely being limited): About constant drug concentrations within the pellets combined with sink conditions outside the pellets are expected to result in about constant release rates (given the condition, that the film coatings do not change over time).

Furthermore, it can be seen that in certain cases (especially for the thicker film coatings and in phosphate buffer pH = 7.4), more or less important lag times were observed prior to the onset of drug release. 

This might serve as an indication that crack formation is eventually of importance, especially because the length of the lag time is strongly pellet dependent. An alternative explanation might be that it takes time for the water to penetrate into the pellets, dissolve the drug and for the first drug molecules to diffuse out through the initially drug-free film coating. However, if this was true, more homogeneous lag times could be expected, since these phenomena likely occur at the same rates in all pellets. However, crack formation in polymeric film coatings can be highly variable from pellet to pellet.

To evaluate the hypothesis of crack formation in the film coating during drug release, also dynamic changes in the size of single pellets upon exposure to the release media were monitored. The pellets were individually placed into flasks containing 500 µL release medium. At predetermined time points, pictures were taken with an optical imaging system. Figure 5 shows the results obtained with pellets coated with 5% Kollicoat SR:IR 95:5 (left hand side) and 10% Kollicoat SR:IR 90:10 (right hand side) upon exposure to acetate buffer pH = 3.5 (top row) or phosphate buffer pH = 7.4 (bottom row). 

As can be seen, the size of certain pellets continuously increases over time, until potentially a plateau is reached or subsequent shrinking is observed. This is a further indication for crack formation: Upon penetration into the system, a hydrostatic pressure is generated within the pellet cores, which acts against the film coating and causes system swelling. Once a crack is formed, the increase in pellet size levels off. In case an important hydrostatic pressure has built up, parts of the core contents are expelled and the system shrinks. Interestingly, in many cases, such leveling off of system swelling is already seen at early time points, which would correspond to early crack formation in many individual pellets. Though few single pellets show a later onset in shrinking or remain in a state of increased size.

### 3.3. Theoretical Prediction of Drug Release

To evaluate the hypothesis of crack formation in the film coating during drug release, an appropriate mathematical model was used to theoretically predict the resulting verapamil release kinetics in acetate buffer pH = 3.5 in the absence of crack formation. The following equation can be derived from Fick’s law of diffusion under the given initial and boundary conditions [44]:(6)$$M_{t}=4\cdot \pi \cdot R_{o}\cdot R_{i}\cdot \left(R_{o}-R_{i}\right)\cdot C_{s}\cdot K\cdot \left[\frac{D\cdot t}{{\left(R_{o}-R_{i}\right)}^{2}}-\frac{1}{6}-\frac{2}{\pi^{2}}\cdot \overset{\infty }{\underset{n=1}{\sum }}\frac{{\left(-1\right)}^{n}}{n^{2}}\cdot exp\left\{-\frac{D\cdot n^{2}\cdot \pi^{2}\cdot t}{{\left(R_{o}-R_{i}\right)}^{2}}\right\}\right]$$
where, *M_t_* denotes the absolute cumulative amount of drug released at time *t*; *R_i_* and *R_o_* the inner and outer radii of the coated pellets; *D* is the apparent diffusion coefficient of the drug in the polymeric film coating; *K* represents the partition coefficient of the drug between the film coating and the wetted pellet core, and n is a dummy variable. This equation considers:the spherical geometry of the pellets,initially drug free film coatings, causing a lag time prior to drug release,verapamil portioning from the wetted pellet core into the polymeric film coating,drug diffusion through the Kollicoat SR:IR coatings that is much slower than water penetration into the system and drug dissolution in the pellet core,sink conditions in the surrounding bulk fluid,constant concentrations of dissolved drug in the pellets’ cores (hence, saturated drug solutions),time-independent diffusion coefficients through the film coatings,constant film coatings thicknesses,constant drug concentrations inside the pellet core (please note that the amounts of water available for drug dissolution are limited inside the pellets, at least at early time points).

The outer radius of the coated pellets was determined by optical microscopy. Considering a density of 1 g/mL, the thicknesses of the polymer layers were estimated to be equal to 7.6 and 14.9 µm in the case of the Kollicoat SR:IR 95:5 and 90:10 blends, respectively. The partition coefficients of verapamil HCl between the wetted pellet cores and the Kollicoat SR:IR film coatings were estimated as follows: Thin polymeric films of identical composition were exposed to acetate buffer pH = 3.5 containing 30 or 45 mg/mL verapamil hydrochloride (assuming that this medium can simulate the liquid in the cores). Please note that when exposing the films to saturated drug solutions, they very slowly dissolved. During several days (until equilibrium was reached), film samples were withdrawn and their drug content was determined, as well as the drug content in the surrounding bulk fluid. The following partition coefficients were determined for systems based on Kollicoat SR:IR 95:5 and 90:10 blends (mean values obtained at 30 and 45 mg/mL verapamil hydrochloride): 5.4 ± 1.5 and 5.8 ± 1.7, respectively.

Equation (6) can be used to theoretically predict verapamil release from the Kollicoat SR:IR 95:5 and 90:10 coated pellets, assuming that drug diffusion through the intact polymeric film coating is the dominant mass transport step. Please note that it considers an initial lag phase due to the fact that the film coatings are initially drug free. The model also assumes a constant concentration of dissolved verapamil in the pellet cores, e.g., saturated drug solutions (due to the fact that at least initially the amounts of water available to dissolve the drug are limited). Consequently, the theory does not cover the entire drug release period. The curves in Figure 6 show these theoretical drug release predictions, while the symbols illustrate the independent experimental values. Clearly, the theory underestimates verapamil release at both Kollicoat SR:IR blend ratios. This is consistent with the above described hypothesis of crack formation in the polymeric film coatings, so drug release is not primarily controlled by diffusion through the intact film coating.

### 3.4. SEM and Raman Imaging

The top row of Figure 7 shows SEM pictures of verapamil HCl loaded starter cores coated with 5% Kollicoat SR:IR 95:5 blends (left hand side) or 10% Kollicoat SR:IR 90:10 blends (right hand side) before exposure to the release medium. As it can be seen, no sign indicating the presence of cracks is visible. This is in contrast to pellets observed after 45 min exposure to acetate buffer pH = 3.5 and after 120 min exposure to phosphate buffer pH = 7.4, irrespective of the Kollicoat SR:IR ratio/coating level: In those cases, clear evidence for crack formation was visible, in particular upon exposure to acetate buffer pH = 3.5. This is consistent with the observed verapamil release behavior from ensembles of pellets (Figure 1) as well as from single pellets (Figure 3 and Figure 4): Drug release was faster at low pH, due to the higher solubility of the drug, as discussed above. However, these SEM pictures obtained after exposure to the release media must be seen with great caution, because the pellets were dried before imaging. This drying process introduced artifacts. During drug release the film coatings contain considerable amounts of water. For this reason, also Raman imaging was used to monitor potential changes in the film coatings and pellets cores during drug release.

Figure 8a and Figure 9a show Raman images obtained from cross sections of verapamil HCl loaded starter cores coated with 5% Kollicoat SR:IR 95:5 or 10% Kollicoat SR:IR 90:10 blends, respectively. The cross sections were obtained using a scalpel. As indicated, the pellets were investigated before or after different exposure times to acetate buffer pH = 3.5 or phosphate buffer pH = 7.4. It has previously been shown that the Raman spectra of Kollicoat SR and IR, verapamil HCl and the other components of the pellet starter cores are sufficiently different to allow for reliable imaging [41]. Looking at the top pictures in Figure 8a and Figure 9a, it can be seen that the drug is distributed throughout the pellet cores in the form of tiny particles and particle agglomerates. Importantly, the overall distribution within the spherical cores is homogeneous. Furthermore, the film coating is visible (marked in yellow), surrounding the pellet cores. Importantly, based on this type of images of cross sections of pellets obtained by cutting coated pellets with a scalpel, great caution must be paid with respect to the integrity of the polymeric films: The scalpel introduced artefacts and the absence of yellow color at a certain position around the blue starter cores might lead to the conclusion that the film coating was not continuous upon manufacturing. In the present study, this would be an erroneous conclusion, since non-invasive Raman imaging revealed continuous Kollicoat SR:IR films before exposure to the release media, as illustrated in Figure 8b and Figure 9b: Irrespective of the polymer blend ratio and coating level, continuous films were observed (depicted in yellow).

The other Raman pictures in Figure 8a and Figure 9a show cross-sections (obtained upon cutting with a scalpel) of pellets coated with 5% Kollicoat SR:IR 95:5 or 10% Kollicoat SR:IR 90:10 blends after different time periods of exposure to acetate buffer pH = 3.5 and phosphate buffer pH = 7.4 (as indicated). Clearly, multiple defects are visible in the film coatings. However, based only on these pictures, it remains unclear whether they have been created during drug release or during cutting with the scalpel. The non-invasively obtained Raman images in Figure 8b and Figure 9b allow to clarify this question: Various coating defects can be seen after exposure to the release medium, irrespective of the polymer blend ratio/coating level and type of bulk fluid. Together with the other experimental and theoretical evidence described and discussed above, this is a strong indication for the fact that verapamil is released at least in part through cracks formed in the Kollicoat SR:IR film coatings during drug release.

Please note that the non-invasive Raman imaging (Figure 8b and Figure 9b) also allows monitoring the disappearance of the verapamil HCl particles inside the pellet cores: With increasing exposure time, less drug particles (depicted in red) are visible. Please note that the images only show one (virtual) cross section of the pellets: Even if no drug particles are visible on such an image, this does not mean that the entire pellet core does not contain any drug.

## 4. Conclusions

The present study highlights the importance of the pH of the liquid within the pellets cores and of crack formation in the film coatings for verapamil release from systems coated with different Kollicoat SR:IR blends. It was shown that non-invasive analytical techniques, such as monitoring of dynamic changes in the pH inside film-coated pellets during drug release using fluorescence spectroscopy as well as the detection of crack formation by confocal Raman microscopy, can be very helpful to better understand the underlying drug release mechanisms from polymer coated pellets. In contrast to invasive analytical tools, the risk of artefact creation can be substantially reduced. SEM imaging and invasive cross-sectioning of pellets confirmed findings on drug release mechanisms through cracks and showed that the weakly basic drug was rapidly released in the case of exposure to the slightly acidic dissolution medium.

## Figures and Tables

**Figure 1 pharmaceutics-13-01723-f001:**
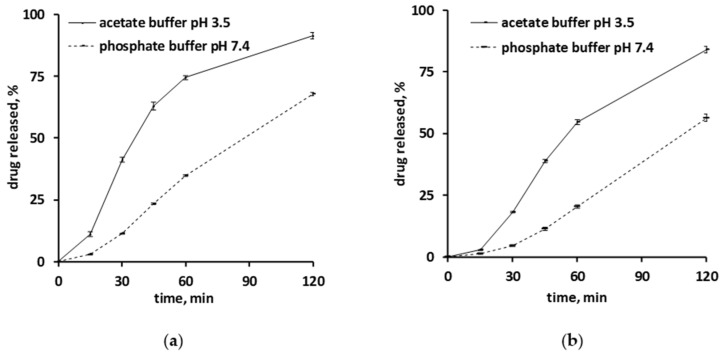
Verapamil release in acetate buffer pH = 3.5 and phosphate buffer pH = 7.4 from pellets coated with: (**a**) Kollicoat SR:IR 95:05 (coating level: 5%), or (**b**) Kollicoat SR:IR 90:10 (coating level: 10%).

**Figure 2 pharmaceutics-13-01723-f002:**
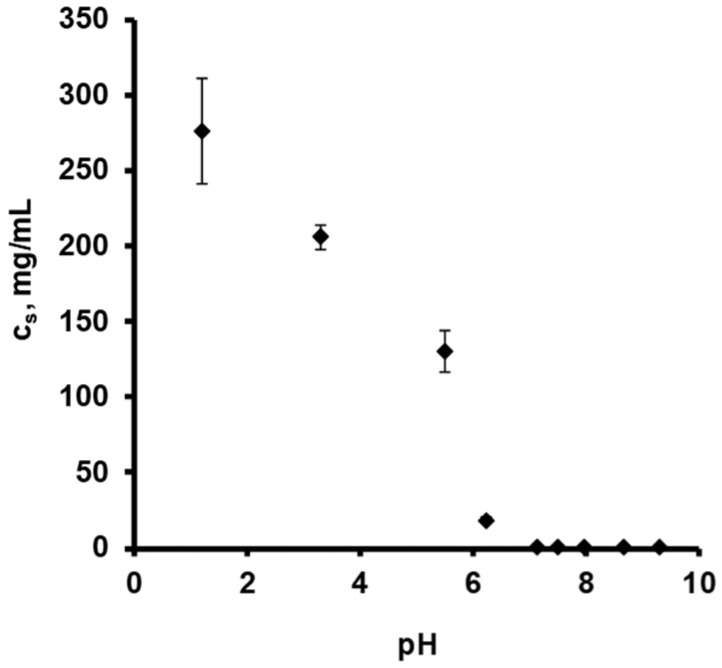
Solubility (c_s_) of verapamil HCl at 37 °C, as a function of the pH in different aqueous release media: 0.1 N HCl, acetate buffer pH = 3.5 and phosphate buffer pH = 7.4. The pH was optionally adjusted with varying amounts of sodium hydroxide or HCl.

**Figure 3 pharmaceutics-13-01723-f003:**
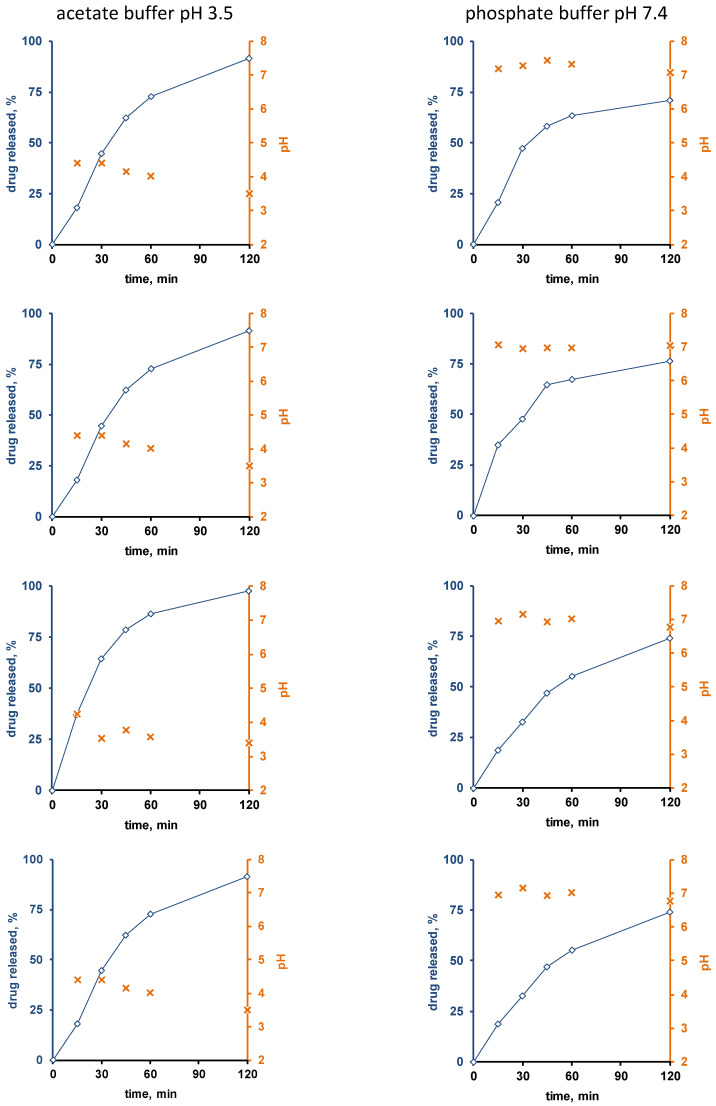
Drug release (blue curves, right y-axes) from single pellets coated with Kollicoat SR:IR 95:05 (coating level: 5%) upon exposure to acetate buffer pH = 3.5 (**left column**) or phosphate buffer pH = 7.4 (**right column**). The orange curves (left y-axes) show the pH values determined by fluorescence spectroscopy inside the pellet cores. Each diagram shows the drug release from/pH changes within the same single pellet.

**Figure 4 pharmaceutics-13-01723-f004:**
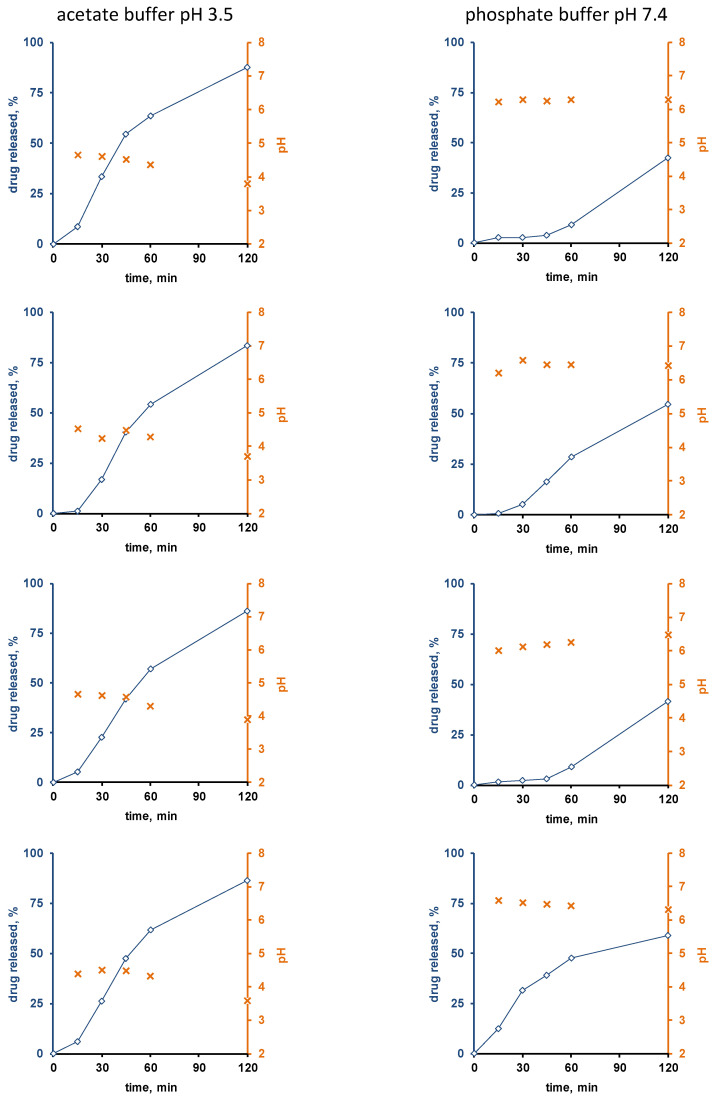
Drug release (blue curves, right y-axes) from single pellets coated with Kollicoat SR:IR 90:10 (coating level: 10%) upon exposure to acetate buffer pH = 3.5 (**left column**) or phosphate buffer pH = 7.4 (**right column**). The orange curves (left y-axes) show the pH values determined by fluorescence spectroscopy inside the pellet cores. Each diagram shows the drug release from/pH changes within the same single pellet.

**Figure 5 pharmaceutics-13-01723-f005:**
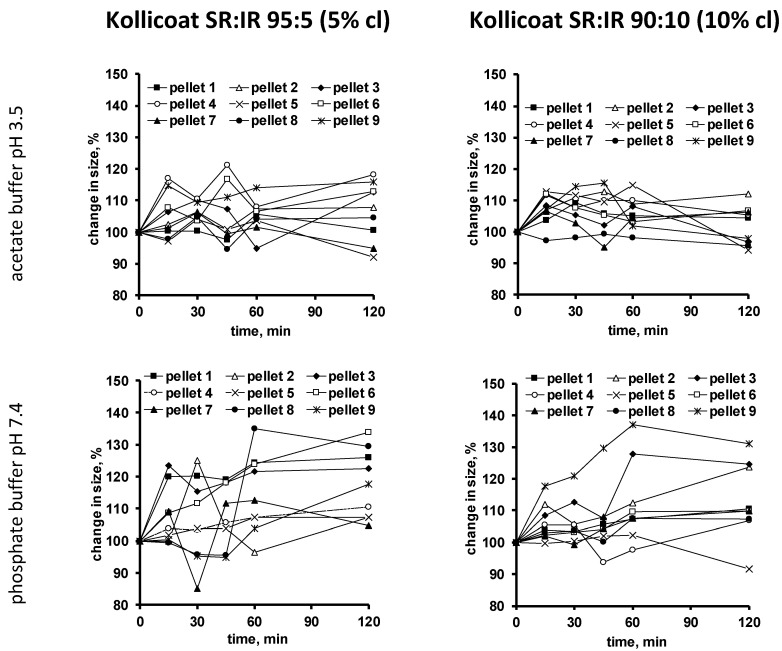
Dynamic changes in the size of single pellets coated with Kollicoat SR:IR 95:5 (5% coating level (cl), left hand side) or Kollicoat SR:IR 90:10 (10% coating level, right hand side) upon exposure to acetate buffer pH = 3.5 (**top row**) or phosphate buffer pH = 7.4 (**bottom row**).

**Figure 6 pharmaceutics-13-01723-f006:**
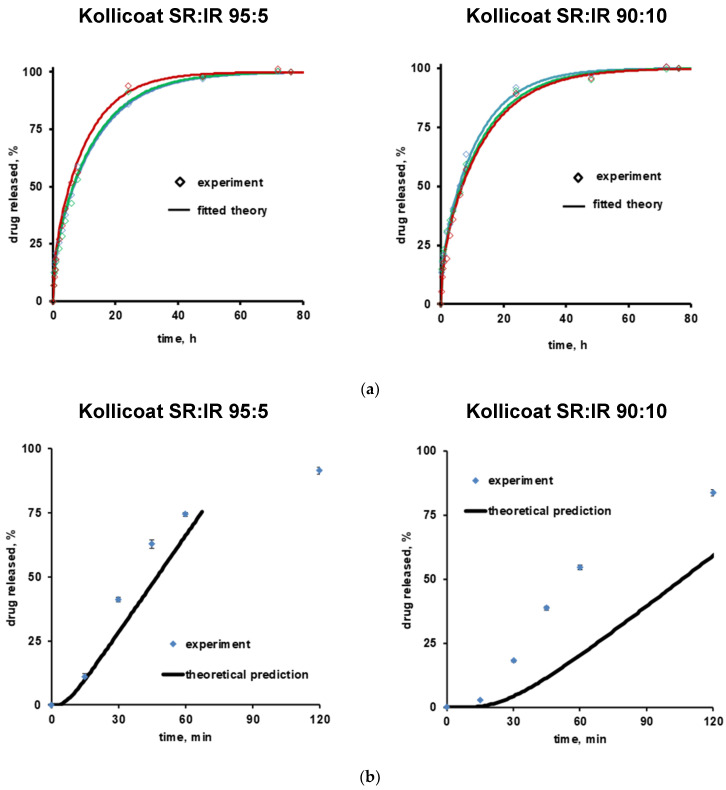
Theory (curves) and experiments (symbols): (**a**) Drug release from free verapamil HCl-loaded films in acetate buffer pH = 3.5 (initial drug loading: 1%, *w/w*). The curves show the fittings of Equation (1). Drug release from 3 individual films is shown (red, blue and green). (**b**) Drug release from verapamil HCl-loaded, coated pellets. The curves show the theoretical predictions obtained using Equation (6). The free films and film coatings were based on Kollicoat SR:IR 95:5 or 90:10 blends. The film thickness was about 100 µm in the case of free films, and about 8 and 15 µm in the case of the film coatings based on Kollicoat SR:IR 95:5 and 90:10 blends, respectively (corresponding to 5% and 10% coating level).

**Figure 7 pharmaceutics-13-01723-f007:**
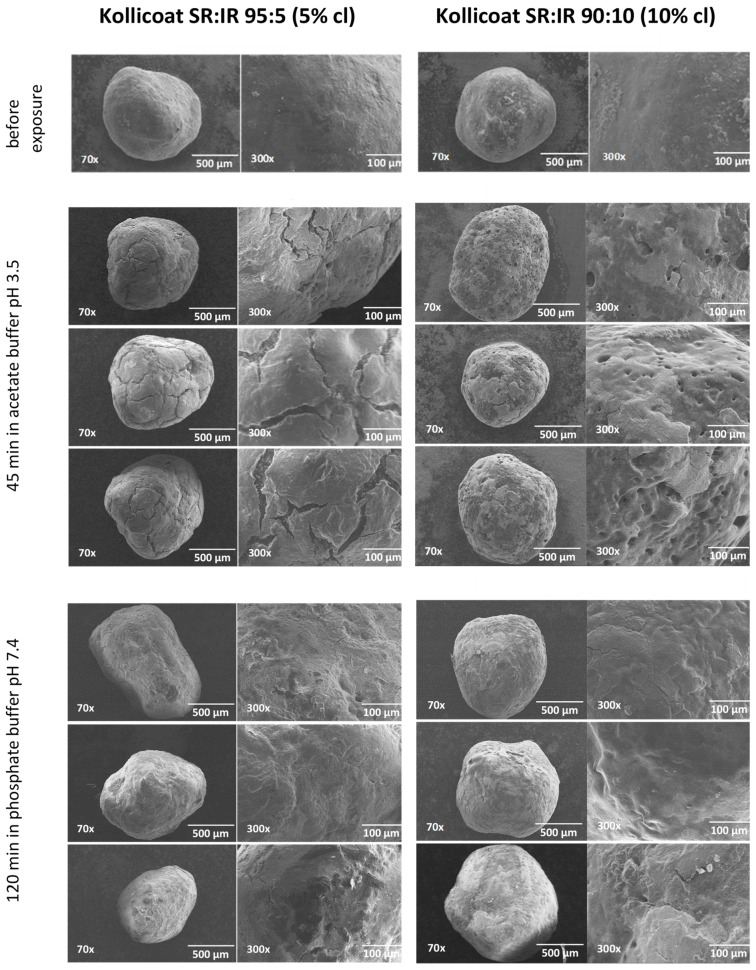
SEM images of pellets coated with Kollicoat SR:IR 95:5 (5% coating level, left column) or Kollicoat SR:IR 90:10 (10% coating level, right column) before and after exposure to acetate buffer pH = 3.5 or phosphate buffer pH = 7.4 (as indicated).

**Figure 8 pharmaceutics-13-01723-f008:**
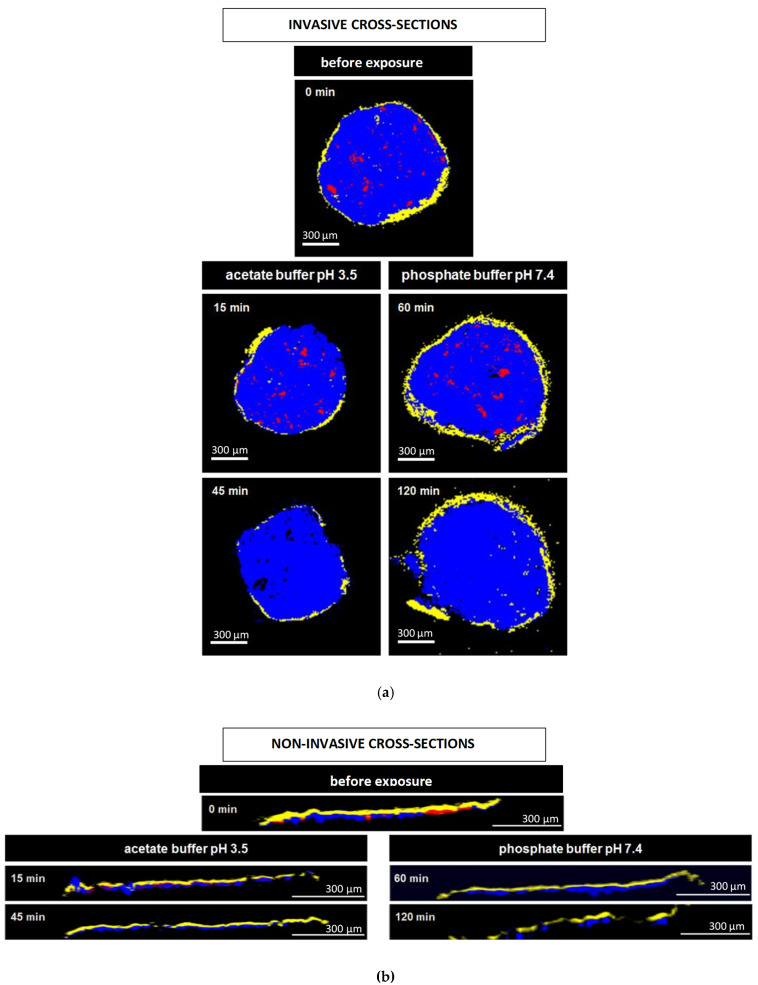
Raman images of cross sections of pellets coated with 5% Kollicoat SR:IR 95:5 after exposure to different release media for varying time periods (as indicated). The cross sections were obtained: (**a**) invasively (by cutting with a scalpel), or (**b**) non-invasively (virtually by confocal Raman spectroscopy). False colors depict the matrix in blue, the film coating in yellow and the drug in red.

**Figure 9 pharmaceutics-13-01723-f009:**
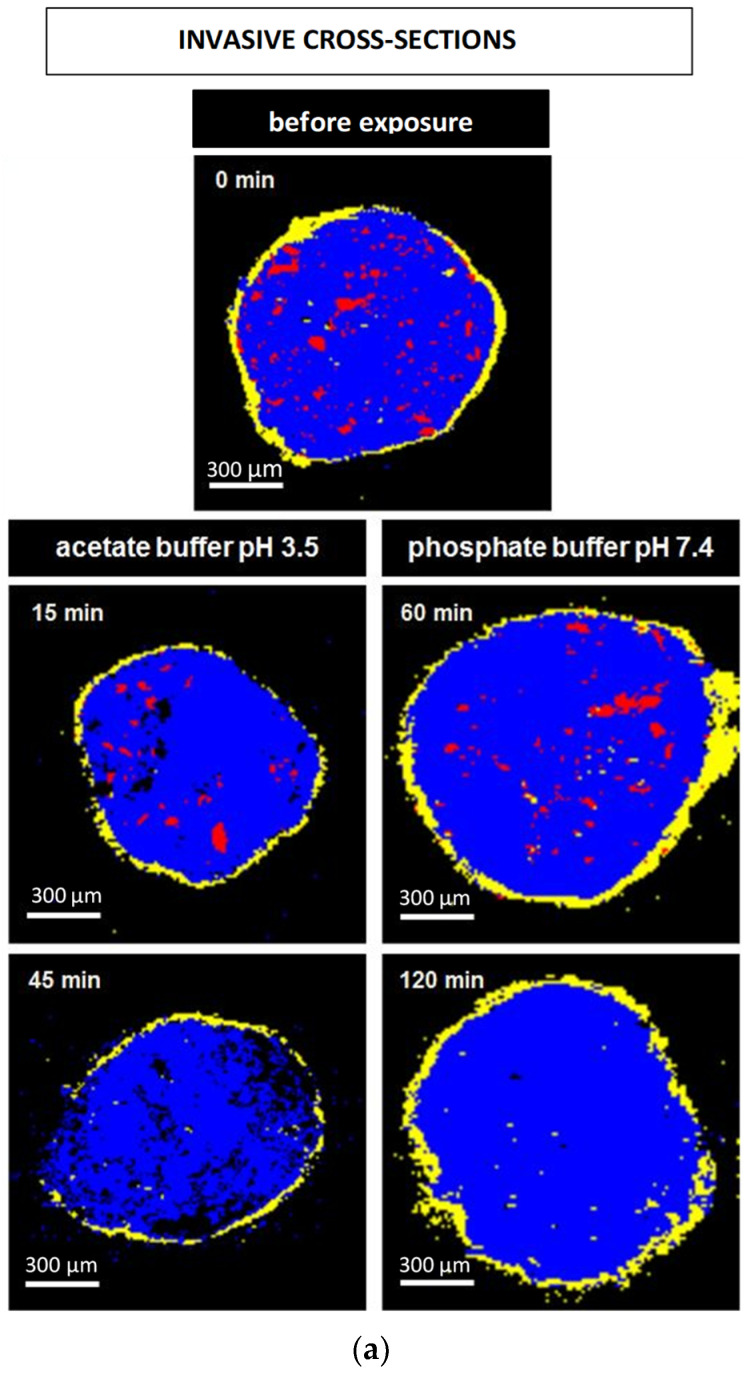
Raman images of pellets coated with 10% Kollicoat SR:IR 90:10 before and after exposure to different release media for varying time periods (as indicated): (**a**) cross sections (obtained invasively by cutting with a scalpel), (**b**) surfaces and virtual cross sections of the film coatings (obtained non-invasively by confocal Raman spectroscopy). False colors depict the matrix in blue, the film coating in yellow and the drug in red.

## Data Availability

The data presented in this study are available on request from the corresponding author.

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
