# Peer review of "Towards a Better Understanding of Verapamil Release from Kollicoat SR:IR Coated Pellets Using Non-Invasive Analytical Tools"

_pharmaceutics, 2021, doi:10.3390/pharmaceutics13101723_

Round 1

Reviewer 1 Report

The manuscript is focusing on a non-invasive analytical solution to monitor how the pH changes inside the pellet during dissolution studies and investigated the integrity of the coating and the creation of cracks using a confocal Raman microscope. Studying these results may help to understand the mechanisms of drug release. The figures are informative; they demonstrate the essential results of the work. The manuscript is generally well-written and presents new and useful contexts on the subject's practical importance. Some small notes: In the case of pellet production, the batch size and the diameter of the nozzle used in the coating process are not indicated.

Author Response

The manuscript is focusing on a non-invasive analytical solution to monitor how the pH changes inside the pellet during dissolution studies and investigated the integrity of the coating and the creation of cracks using a confocal Raman microscope. Studying these results may help to understand the mechanisms of drug release. The figures are informative; they demonstrate the essential results of the work. The manuscript is generally well-written and presents new and useful contexts on the subject's practical importance. Some small notes: In the case of pellet production, the batch size and the diameter of the nozzle used in the coating process are not indicated.

Reply: Minor spelling errors have been corrected and information on pellet starter core production and the coating process added.

Reviewer 2 Report

The authors described all their experiments in great detail. In my opinion, it could be interesting and inspiring for other researchers in the field of pellet technology and other modern forms of oral medications. To me the most interesting aspect is related to non-invasive monitoring the local pH within the pellet cores during drug release, using fluorescence spectroscopy. In this way, the importance of the pH of the liquid in the pellet cores was shown.

The strength of this article is also to draw attention to the risk of artefact creation and to show some possibilities to avoid them.

Author Response

The authors described all their experiments in great detail. In my opinion, it could be interesting and inspiring for other researchers in the field of pellet technology and other modern forms of oral medications. To me the most interesting aspect is related to non-invasive monitoring the local pH within the pellet cores during drug release, using fluorescence spectroscopy. In this way, the importance of the pH of the liquid in the pellet cores was shown.

The strength of this article is also to draw attention to the risk of artefact creation and to show some possibilities to avoid them.

Reply: Minor spelling errors have been corrected.

Reviewer 3 Report

The manuscript titled “Towards a better understanding of verapamil release from 2 Kollicoat SR:IR coated pellets using non-invasive analytical 3 tools” describes the use of different analytical tools to examine the verapamil release from coated pellets. The work is novel and displays valuable results however some parts of the manuscript were poorly written, undervaluing it. Also, some experimental designs require revision to improve the quality of the data derived.

The introduction lacks focus and does not flow properly. Using the term “interesting” does not make a good justification and is not a reason to explain the rationale of this work. Please consider re-writing the introduction to better explain the aims of this work.

It is not clear in the manuscript why acetate buffer (pH 3.5) and phosphate buffer (pH 7.4) were chosen. Are either of these media really physiologically relevant? IS this based on pharmacopeial recommendation? If this is aa 2 hr release, why wasn’t 0.1M HCl (pH 1.2) used?

More detailed comments:

Line 59: replace the colon with a do after “at the same time”.

Line 77: strong not strongly

Section 2.2: volumes used? Number of samples used? Weight of verapamil used?

Section 2.5: what weight or number of ensemble pellets was used? Number of repetitions? Were sinkers used? How did you ensure that the pellets did not float?

Figure 1, 3, 4: where are the error bars?

Page 11: how is 500 μL physiologically relevant? Whether a single pellet or ensemble, the physiological volumes it/they are exposed to is not in the μL range.

Figure 5: there is a clearly high variation here; what could the reason(s) be?

Figures 8 and 9: it is clear from the images that the coating thickness is not equal on the surface of the pellet; could this explain the difference in the swelling behavior?

The discussion and conclusions require expanding as they currently both lacking.

Author Response

The manuscript titled “Towards a better understanding of verapamil release from 2 Kollicoat SR:IR coated pellets using non-invasive analytical 3 tools” describes the use of different analytical tools to examine the verapamil release from coated pellets. The work is novel and displays valuable results however some parts of the manuscript were poorly written, undervaluing it. Also, some experimental designs require revision to improve the quality of the data derived.

The introduction lacks focus and does not flow properly. Using the term “interesting” does not make a good justification and is not a reason to explain the rationale of this work. Please consider re-writing the introduction to better explain the aims of this work.

Reply: Adjustments have been performed to this part of the manuscript.

It is not clear in the manuscript why acetate buffer (pH 3.5) and phosphate buffer (pH 7.4) were chosen. Are either of these media really physiologically relevant? IS this based on pharmacopeial recommendation? If this is aa 2 hr release, why wasn’t 0.1M HCl (pH 1.2) used?

 Reply: The acetate buffer had been chosen to be within the detection range of the pH probe Oregon Green. Drug release in 0.1 N HCl has also been performed and did not reveal significant differences towards pH 3.5 (data not shown).

More detailed comments:

Line 59: replace the colon with a do after “at the same time”.

Reply: The line has been rephrased.

Line 77: strong not strongly

Reply: Your remark has been taken into consideration.

Section 2.2: volumes used? Number of samples used? Weight of verapamil used?

Reply: The volumes & samples used have been added to section 2.2. Since the drug’s solubility varied with the differing pH values and excess amounts of verapamil HCl have been added, the weight of drug within each individual flask was different.

Section 2.5: what weight or number of ensemble pellets was used? Number of repetitions? Were sinkers used? How did you ensure that the pellets did not float?

Reply: The mass of pellet samples had been added. During the dissolution studies the drug loaded pellets did not exhibit floating.

Figure 1, 3, 4: where are the error bars?

Reply: The error bars are very well present in Figure 1, they do vary from 0 to 1.58%, 0 to 0.66% for the 95:5 blend (5% coating level), 0 to 1.28% and 0 to 1.69% for the 90:10 blend (10% coating level) in acidic and neutral medium, respectively (they are simply hardly visible). Concerning the Figures 3 and 4, please note, that drug release and pH values shown here are obtained from single (individual) pellets, and do not represent mean values.

Page 11: how is 500 μL physiologically relevant? Whether a single pellet or ensemble, the physiological volumes it/they are exposed to is not in the μL range.

Reply: It has to be considered, that individual (single) pellets are not able to deliver a pharmacologically relevant dose. Indeed, here very small amounts of dosage forms are exposed to very small amounts of dissolution medium.

Figure 5: there is a clearly high variation here; what could the reason(s) be?

Reply: The pellets behave very differently when observed individually, as reported by Marucci, M.; Ragnarsson, G.; Nilsson, B.; Axelsson, A. (Journal of Controlled Release 2010, 142, 53–60, doi:10.1016/j.jconrel.2009.10.009). The paragraph had been rephrased.

Figures 8 and 9: it is clear from the images that the coating thickness is not equal on the surface of the pellet; could this explain the difference in the swelling behavior?

 Reply: Please note that cross-sections shown in Figure 8a & 9a have been performed invasively and are the reason for partial displacement of the coating layer (here the core composition is placed in the focus of the reader). However, the pellet cores have not been perfectly spherical, resulting in very slight coating thickness variations.

The discussion and conclusions require expanding as they currently both lacking.

Reply: The discussion section has been revised. The conclusion paragraph reflects a rationale highlighting the novelties represented in the manuscript. Not all aspects of the manuscript are represented here, since certain results served to validate findings obtained by the non-invasive techniques.

Round 2

Reviewer 3 Report

I can't see any improvements/changes to the introduction, discussion or conclusions. They still exactly the same with only grammatical corrections made. I still believe these sections require major improvement. In its current form, there no clear background story to explain the rationale of this study. It is not clear for all readers, especially non-expert readers or those from other backgrounds.

With regards to the error bars of figure 1, these can't be seen due to the large marker sizes. Either reduce the marker size to make the error bars visible or completely remove the markers and differentiate the lines by having 2 different line styles (e.g., solid and dashed).

My other comments have been addressed/corrected.

Author Response

I can't see any improvements/changes to the introduction, discussion or conclusions. They still exactly the same with only grammatical corrections made. I still believe these sections require major improvement. In its current form, there no clear background story to explain the rationale of this study. It is not clear for all readers, especially non-expert readers or those from other backgrounds.

Reply: The introduction & discussion has been reformulated and specific terms have been vulgarized to guide the reader towards the essential challenges when characterizing coated pellet formulations. However, these revisions have been made with respect to the original text (manuscript after 1st revision) which had been approved by reviewer #1 & #2. The conclusion has been rephrased and terms of general comprehension have been included.

With regards to the error bars of figure 1, these can't be seen due to the large marker sizes. Either reduce the marker size to make the error bars visible or completely remove the markers and differentiate the lines by having 2 different line styles (e.g., solid and dashed).

Reply: Your request has been addressed, the markers have been reduced.

My other comments have been addressed/corrected.